# Application of a Caries Treatment Difficulty Assessment System in Dental Caries Management

**DOI:** 10.3390/ijerph192114069

**Published:** 2022-10-28

**Authors:** Yu Wei, Jingqian Wang, Dongyue Dai, Haohao Wang, Min Zhang, Zhigang Zhang, Xuedong Zhou, Libang He, Lei Cheng

**Affiliations:** 1State Key Laboratory of Oral Diseases, National Clinical Research Center for Oral Diseases, Department of Operative Dentistry and Endodontics, West China School of Stomatology, Sichuan University, Chengdu 610041, China; 2Department of Stomatology, The People’s Hospital of Dazu, Chongqing 402360, China

**Keywords:** caries risk assessment, treatment difficulty assessment, dental caries management, software

## Abstract

Dental caries is one of the most common chronic diseases caused by progressive bacteria, affecting all age groups. Today, restorative fillings are widely used for dental caries treatment, but the restorative treatment has a high failure rate. Meanwhile, many researchers have discovered the differences of caries risk among populations by using the caries risk assessment and put forward a new standpoint that caries should be treated individually. Therefore, our research group established a Dental Caries Treatment Difficulty Assessment system in a previous study. This time, we combined the caries risk assessment with the caries treatment difficulty assessment, then used Python to design a Dental Caries Management Software. The purpose of this case report is to present a case applying this software in dental caries management and other data collected in Chengdu, China, with this software on the assessment of caries treatment difficulty. Patients with personalized assessment and management can achieve good treatment results, including reducing the risk and treatment difficulty of dental caries. At the same time, other cases show that the software has good application potential in individual management and group information collection. These cases indicate that the software enables dentists to carry out both the risk assessments and the treatment difficulty assessment of patients, and it has the potential as a tool for epidemiological investigation. It also enables dentists and patients to have a basic understanding of the dental health status of patients and create personalized dental caries treatment, so as to achieve the goal of controlling the progression of dental caries and rebuilding the structure and restoring the function of teeth.

## 1. Introduction

Dental caries is a dynamic, preventable, reversible and complex biofilm-mediated multi factorial disease [1], which is the most prevalent non-communicable disease in the world [2], affecting all age groups throughout the whole life cycle [3]. Dental caries causes a significant personal, social, and economic burden on a global scale [4,5]. About 2.3 billion people have untreated tooth decay in their permanent teeth, and more than 530 million children have untreated tooth decay in their primary teeth [6,7]. While tooth decay is largely preventable [8], untreated cavities in permanent teeth ranked first in prevalence of all 291 diseases and injuries in the entire Global Burden of Disease Study [9,10]. Although with the development of science and technology there are now varieties of methods to treat dental caries [11,12], the failure rate of dental caries treatment remains high [13].

Researchers observed 5493 failures in 27,407 dental restorative fillings. Meanwhile, the median lasting time of dental restorative fillings was only 207 months. The annual failure rates of grade I, II, III, IV and V restorations were 3.8%, 4.0%, 4.6%, 4.9% and 3.9%, respectively. Several factors, such as practice level (dental skill), patient physiological condition (age) and dental condition (restoration material, restoration morphology according to the caries classification provided by Dr. G. V. Black) contribute to the failure rate [14]. Therefore, according to the specific situation of caries, the key to the treatment of caries is the individualized management of dental caries. However, the current clinical caries management fails to do so.

To solve this problem, we have proposed the concept of caries-whole-life-cycle management [15] and established a caries treatment difficulty assessment in a previous study [16]. Whole life-cycle caries management requires dentists to carry out the population management of caries according to the physiological characteristics of patients at different ages and carry out the personalized management of caries according to different risk assessment. Meanwhile, the caries treatment difficulty assessment requires dentists to score the difficulty factors in the treatment and complete the treatment difficulty assessment.

With the development of digital society, many new technologies have been applied to the treatment of dental diseases [17,18,19,20]. At the same time, it also brings new challenges and opportunities for dental caries management. Hence, we combed the well-applied caries risk assessment [21] with caries treatment difficulty assessment that we had established previously, then programmed and applied the Dental Caries Management Software. This is the first software that applied caries treatment difficulty assessment. The purpose of this paper is to report on a case that has been treated and managed with this software for five years. The 253 cases collected with this software will also be shown. Patients with personalized assessment and management can achieve good treatment results, including reducing the risk and treatment difficulty of dental caries. At the same time, other cases show that the software has good application potential in individual management and group information collection. These cases show the software that enables dentists to carry out both the risk assessments and the treatment difficulty assessment of patients, and it has the potential as a tool for epidemiological investigation. We would like to prove that the application of this system has a good prospect of being put into use as personalized caries management is important and deserves due attention, recommendation and practice.

## 2. Case Reports

### 2.1. Dental Caries Management Software Programming

We established the assessment of evaluating the difficulty of dental caries treatment, according to systemic and dental factors, individual susceptibility to dental caries, technical sensitivity, previous dental filling experience and other auxiliary factors. According to the degree of treatment difficulty, each factor was scored 1–3, and the comprehensive assessment was divided into Grade I, II and III. Patients bearing the results of both the caries risk assessment and the treatment difficulty assessment would be assigned to dentists graded A, B and C according to their experience and ability [16]. The specific classification rules have been elaborated in our previous study and are also listed in Appendix A.

To make it more convenient to use this assessment, we chose Python to build a cloud platform of Dental Caries Management Software. To keep track of different patients, the software considers the risk degree of patients, the state of each lesion, patient management, clinical management, and monitoring. In general, it can achieve the following functions: 1. The software can synchronize data from multiple devices, such as mobile phones, iPads and computers, which means that the software can be used in the clinic to gather information from individual cases to population distribution. 2. For dentists, the software can provide a collection of patient data, detailed treatment process records, caries risk assessment and treatment difficulty assessment details to facilitate population data analysis. 3. For patients, the software can automatically provide a clear map of the level of caries risk and treatment difficulty, then provide personalized suggestions on caries management, which is conducive to the communication between dentists and patients. To sum up, the expected targets of the Caries Management Software are to reduce the incidence of caries, help individualized caries management, maintain dental health, improve patient satisfaction, and improve the prognosis of caries treatment. The process of using this software is shown in Figure 1. 

### 2.2. A Case of Dental Caries

A 26-year-old male patient came to our clinic because of spontaneous pain in his right posterior tooth for one week. The patient had no systemic disease, and the family history showed no abnormalities. He described his job as a waiter with no exposure to acid substances. However, he loves to drink Coca-Cola and drinks more than 1 L of Coca-Cola every day; he also loves snacks and brushes his teeth less than once a day. These are the main causes of his dental caries. Dental examination found that the patient had poor dental hygiene with I-II° calculus supragingival and subgingival. All his teeth had extensive caries on the cervical region of the buccal and labial surfaces. The caries of teeth #14–24 even invades the tooth cusps. No lesions were found in the palatal and lingual surface. The pulpal surfaces of erosive lesions contained brown-colored, leathery, carious dentin. None of the pulp cavities were involved and the teeth remained asymptomatic on percussion, palpation and cold testing, except for tooth #46. The patient’s toothache came from tooth #46. There were two visible holes in teeth 46 buccal and occlusal surface, the perforation could be detected in the buccal surface, and the probing pain was obvious. There was no pain on percussion but severe pain on cold testing and heat testing (Figure 2a–d).

According to his examination, the patients suffer from: 1. gingivitis; 2. dental caries (teeth #17–27, #37–47); and 3. chronic pulpitis (tooth #46). Therefore, we made the following treatment plan: 1. periodontal basic treatment and dental health education; 2. root canal therapy of tooth #46; and 3. permanent fillings for all teeth (Table 1).

The patients’ general information was recorded into the Dental Caries Management Software. First, the history of present disease, systemic history and dental examination were recorded into the software. Secondly, a caries risk questionnaire was completed by the patient who had a visible caries decay, adjacent caries, white spot lesions on the smooth surface of teeth and restorations filled within 3 years. The detection of *Streptococcus mutans* (MS) and *Lactobacillus* (LB) in his mouth was either moderate or severe, and most of his teeth surface was covered with visible dental plaque. He liked to drink Coca-Cola and eat snacks frequently, which might be the main etiology of his caries. After the above information was input into the software, the caries risk rating assessed by the software was high level, and relevant suggestions on caries management were given to patients (Figure 3a,b). Then, we evaluated the difficulty of treatment. There were class I–V cavities in his teeth, almost all teeth were severely damaged by decay. Some teeth had a history of restoration or filling failure. The patient had no pharyngeal reflex or dental phobia but had excessive salivation. Taking all these into consideration, we planned to use composite resin restoration/glass ionomer transition repair for posterior teeth and minimally invasive techniques for anterior teeth. After the above information was put into the software, we came to the conclusion that the patient’s treatment difficulty rating was Grade III and recommended referral to a grade C dentist (dental aesthetic specialist and caries clinical specialist) for treatment (Figure 3c). With the cooperation of the dentists, the patient’s initial treatment was satisfactory (Figure 4a–f).

Finally, the patient was followed up for observation. The modified USPHS standard was used to evaluate the quality of restorations in this software. One year after treatment, we found that the aesthetic effect, margin fitness and color matching of the restorations were excellent. The patient had no secondary caries, but we found the poor retention of a restorative filling in tooth #22 (Figure 5a,b). It can be seen that one year after treatment, the patient’s dental health had improved, and we dealt with the problems we found in time (Figure 5c,d). Then we reassessed his caries risk and it dropped from high risk to medium risk. So, we adjusted his caries management (Figure 6a–c).

USPHS was used to evaluate the restorations again 2 years after treatment, this time we found secondary caries in the patient’s teeth (Figure 7). We removed secondary caries and restored the cavity again in time, then he reassessed the caries risk routinely, which was still at medium risk.

However, after this return visit, the patient was lost to follow-up for 3 years. Three years later (5 years after the first treatment), the patient came back with dental plaque covering the whole tooth surface, severe periodontal inflammation, many secondary caries and the retention, margin fitness and color matching of the restorations were poor (Figure 8a). Retreatment at this time could not achieve a perfect therapeutic effect (Figure 8b) and the reassessment of his caries risk this time was high risk again.

To sum up, using the caries risk assessment, patients received personalized caries management plans. According to the caries treatment difficulty assessment, patients received appropriate referrals and treatments. The effect of the first treatment was satisfied, which was a good foundation for later maintenance. Regular follow-up visits enable dentists to deal with new problems in time. However, this case suggests that the loss of follow-up in just 3 years almost destroyed the dentist’s previous efforts, and even careful retreatment failed to achieve aesthetic restoration. Therefore, regular follow-up and personalized management of high/medium risk groups is necessary for the treatment and maintenance of dental caries.

### 2.3. Application of Caries Treatment Difficulty Assessment in Population

We now present 253 collected cases with newly diagnosed dental caries who visited dental clinics in Chengdu, China, during 2018–2019. These patients included 176 females (69.6%) and 77 males (30.4%), whose ages ranged from 12 to 71 (the average age was 31.3). In these cases, we found a total of 31 cases categorized into Grade I treatment difficulty, 179 cases into Grade II treatment difficulty and 43 cases Grade III treatment difficulty. There were 50 low-risk patients, 162 medium-risk patients and 41 high-risk patients. According to data above, the highest proportion was found in patients at medium risk of caries with Grade II treatment difficulty (Figure 9).

Robert et al. reported that risk factors for dental caries include three aspects: personal factors, oral environmental factors and factors that directly contribute to caries development. Personal factors include income, sociodemographic status, education and attitude. Oral environmental factors include fluoride, sugars, microbial species and saliva [10]. Meanwhile, the distribution of risk grade and treatment difficulty grade differed slightly among different genders, ages, occupations and several lifestyle habits. According to the Lancet, these indicators may influence personal factors and oral environment.

In terms of gender, the incidence of a high risk of caries and Grade III difficulty of treatment was slightly higher in females than in males (Figure 10a,b). In terms of age, patients with Grade I difficulty and low risk of dental caries tend to decrease with age when they are younger than 59 years old, while those with Grade III difficulty and high risk of dental caries are more likely to be middle-aged people aged 45–59 years old (Figure 10c,d). According to the Occupational Classification Of the People’s Republic of China, Chinese occupations are divided into eight occupational groups (Appendix A) [22]. The patients we included mainly had occupations in the first to sixth categories, with the first and second categories being the dominating ones (Figure 10e). Low-risk, Grade I treatment difficulty was more likely to be found in category 6 occupations, while high-risk, Grade III treatment difficulty was more commonly seen in category 1, 2 and 4 occupations (Figure 10f,g).

The software can collect not only basic information about the patient but also detailed habits that affect the risk of caries and the difficulty of treatment. Therefore, we analyzed some of the patients’ living habits that affect oral environment, hoping to provide some enlightenment for patients’ daily dental health management. According to the chart, we found that those patients who preferred carbonated drinks and sweets had a history of orthodontics and brushed their teeth once or less per day without regular dental examinations were more likely to be rated as medium or high risk, Grade II or Grade III treatment difficulty (Figure 11a–e).

## 3. Discussion

Dental caries is a chronic disease with high incidence and wide impact. However, because of its high rate of treatment failure, how to improve the long-term effectiveness of treatment has been a dilemma for dentists [23]. Great efforts have been made to solve this problem. Some researchers have invented new techniques for detecting caries lesions [24], such as quantitative light-induced fluorescence (QLF) [25] and laser detection [26]. Some have developed a variety of filling materials with long-term antibacterial ability [27,28]. Still some have used probiotics to regulate oral microbiota balance [29], and others have even begun to study caries vaccines [30]. However, in addition to improved methods and materials, standardized caries management is particularly important.

The four-factor theory of caries [31] corresponds to factors that directly contribute to caries development. It mainly refers to the bacteria in biofilm, diet, tooth and time [1,32]. Therefore, dental caries cannot be well controlled only by the effort of the dentist. It is a common phenomenon that the speed of caries treatment cannot keep up with the speed of new dental caries. The management of caries still needs to be improved. Dentists need to improve and standardize the caries treatment process, and patients need to cooperate with the treatment process and daily dental health management. Based on the caries-whole-life-cycle management and caries treatment difficulty assessment system proposed previously, we combined the caries risk system and caries treatment difficulty assessment system to apply a caries management software. There are many caries management software in the world, such as the International Caries Detection and Assessment System (ICDAS), the International Caries Classification and Management System (ICCMS™) and so on, and they are widely used as standardized and reliable caries detection criteria but they only include caries risk assessments. Our software applies not only a caries risk assessment but also a caries treatment difficulty assessment. This software can collect patients’ information, diagnosis and treatment information; assess caries risk and the caries treatment difficulty of patients; and give referral recommendations and daily management suggestions, so as to establish personalized caries management. The software makes it possible for dentists and patients to work together to combat caries.

The case we present is almost at the highest level of caries treatment difficulty assessment. Although the patient had obvious dental caries in his mouth, he did not come to the clinic until he had a toothache. The patient had a negative attitude of keeping up with dental health. For patients such as this, personalized caries management is particularly important. The patient in question was included in the caries management system and was evaluated as the highest caries risk and caries treatment difficulty grade. This patient was assigned to a Grade C dentist on referral advice. After the treatment plan was created, the patient was treated with conventional permanent resin-filled repair. The filling and aesthetic effects of the first treatment were satisfactory. The dentists did their best. At the same time, we also provided him with detailed dental health education and told him to return to the clinic as planned. During the first two return visits, the patient’s dental health was much better than before. Although there were some new problems, the dentist was able to deal with them promptly. A reassessment at the return visit also showed progress in caries risk and caries treatment difficulty grade. The software reformulated a personalized management plan for the patient, further facilitating the efforts of the dentist and the patient. However, this patient was lost to follow-up for 3 years. At this point, we found that all the previous efforts had been almost entirely in vain, and the patient said that he had not followed a personalized caries management plan to maintain his dental health. The reassessment was back to where it started or even worse than before. We can learn from this case that correct assessment, personalized management plans and timely return visits are inseparable conditions to achieve the ideal long-lasting treatment effect.

In addition to managing individual cases, our software can also analyze the information collected from the population. According to the data above, the highest proportion was found in patients at medium risk of caries with Grade II treatment difficulty. This is consistent with the conclusion of previous studies about caries risk assessment [33,34]. The total number of female patients was more than that of male patients, which may be because studies show that women paid more attention to their dental health than men and have a higher probability of tending to their dental problems in a timely manner [35,36]. Patients with Grade I difficulty and a low risk of dental caries tend to decrease with age when they are younger than 59 years old, which may be due to the prolongation of time within the four-factor theory of caries. At the same time, we can also analyze the statistics regarding the living habits of patients. Overall, this software can not only be used for the effective management for patients in the clinic but also has the potential to be used in epidemiological investigations.

In general, our goal is to promote the application of the dental caries treatment difficulty assessment, to classify the difficulty of different cases and to achieve the optimized allocation of social resources, despite the situation regarding the lack of dental resources in the world, so that patients can attain more reasonable treatment resources. This software would also allow dentists to obtain a basic understanding of the treatment difficulty assessment of their cases before performing an operation. In addition, the dental health of patients can be understood by dentists and patients themselves, and individualized dental health and disease prevention programs can be developed to achieve the goal of controlling the progression of caries and rebuilding the structure and function of teeth.

## 4. Conclusions

The incidence of caries is high all over the world, and the rate of untreated and therapeutic failure is also high. Based on the “Caries Treatment Difficulty Assessment” we proposed previously, we built a cloud “Dental Caries Management Software” platform using Python. This software can be operated and managed by multiple devices at the same time, automatically evaluate the risk level and treatment difficulty level and has the functions of individual case management and population data collection. In conclusion, we have introduced the caries treatment difficulty assessment system into clinical practice as part of caries management. Combined with caries risk assessment, it can simultaneously guide dentists’ clinical treatment and patients’ dental health management. Our cases show that it has potential application prospects in the management of high-risk and difficult populations and also has the potential as a tool for epidemiological investigation and analysis.

## Figures and Tables

**Figure 1 ijerph-19-14069-f001:**
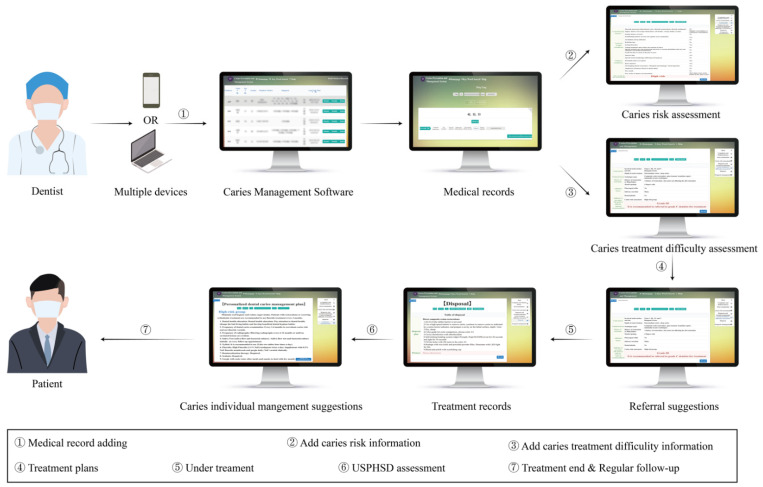
Application process of Caries Prevention and Management Software.

**Figure 2 ijerph-19-14069-f002:**
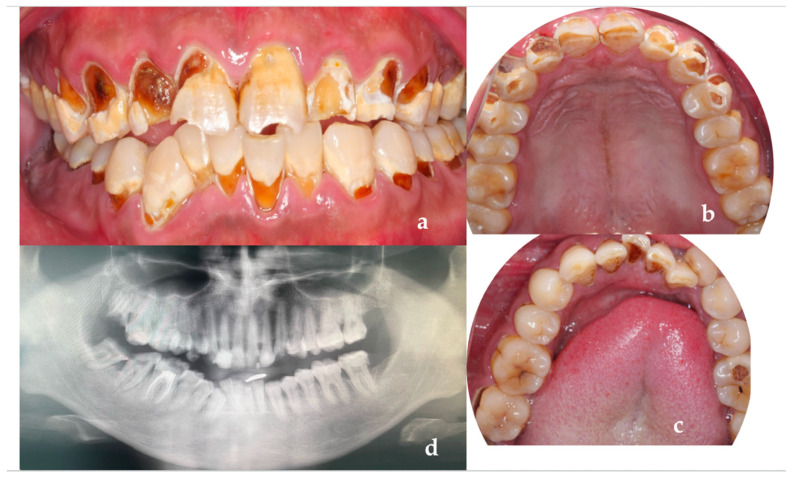
(**a**) The photograph of the patient at first visit; (**b**) Upper teeth; (**c**) Lower teeth; (**d**) Panoramic X-ray of the patient (after the root canal therapy of teeth 46).

**Figure 3 ijerph-19-14069-f003:**
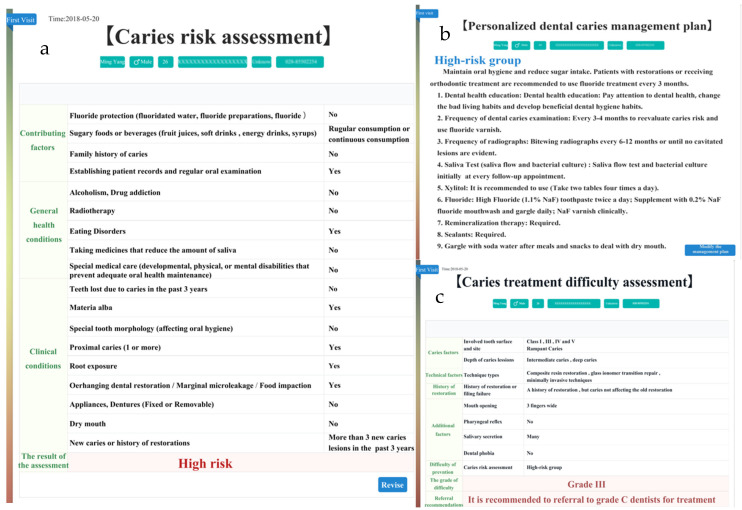
(**a**) Caries Risk Assessment Form; (**b**) Personalized caries management plan; (**c**) Caries treatment difficulty assessment.

**Figure 4 ijerph-19-14069-f004:**
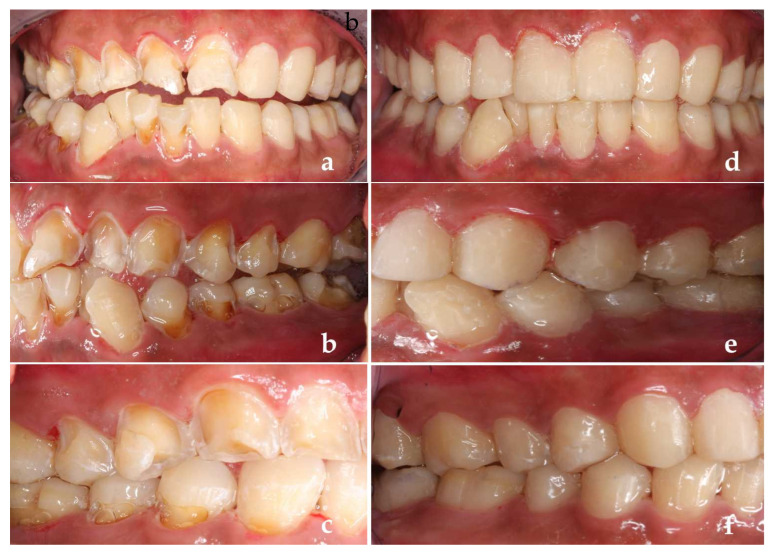
(**a**–**c**) During the first treatment; (**d**–**f**) First treatment completed.

**Figure 5 ijerph-19-14069-f005:**
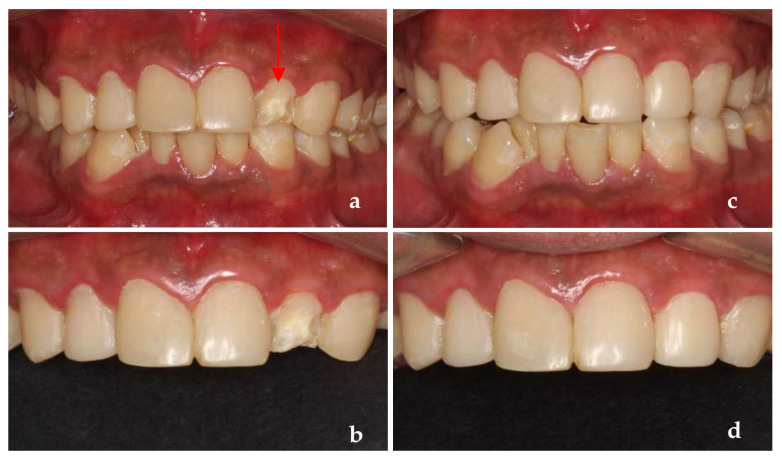
(**a**,**b**) One year after treatment; (**c**,**d**) After the dentist dealt with the new problem in time.

**Figure 6 ijerph-19-14069-f006:**
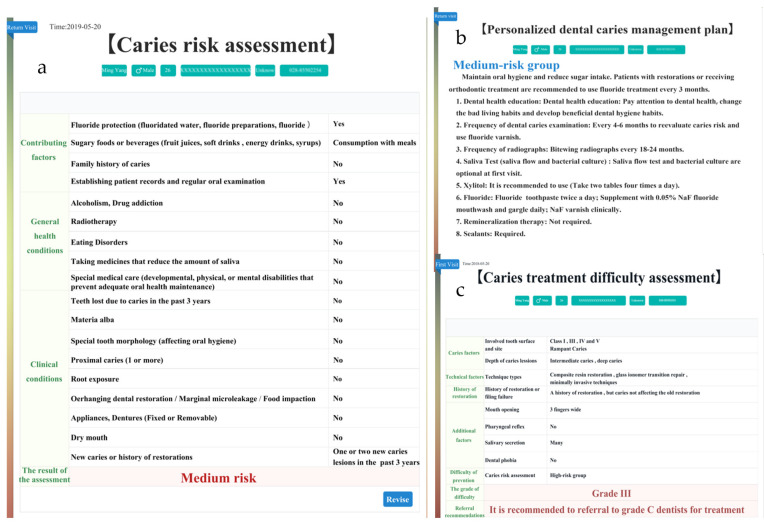
(**a**) Caries Risk Assessment Form of return visit; (**b**) Personalized caries management plan of return visit; (**c**) Caries treatment difficulty assessment of return visit.

**Figure 7 ijerph-19-14069-f007:**
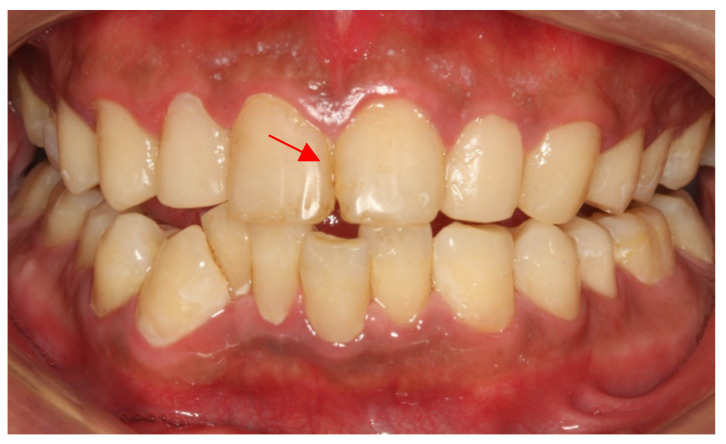
Two years after treatment.

**Figure 8 ijerph-19-14069-f008:**
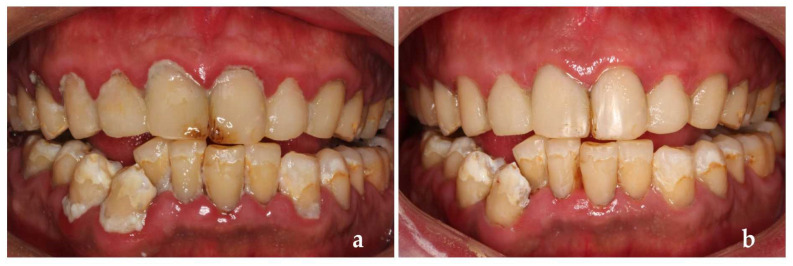
(**a**) Five years after treatment; (**b**) Retreatment completed.

**Figure 9 ijerph-19-14069-f009:**
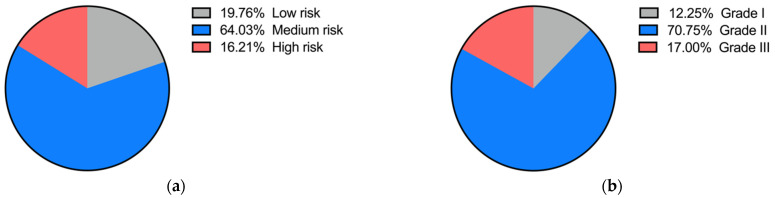
(**a**) Proportion of each caries risk grade; (**b**) Proportion of each treatment difficulty grade.

**Figure 10 ijerph-19-14069-f010:**
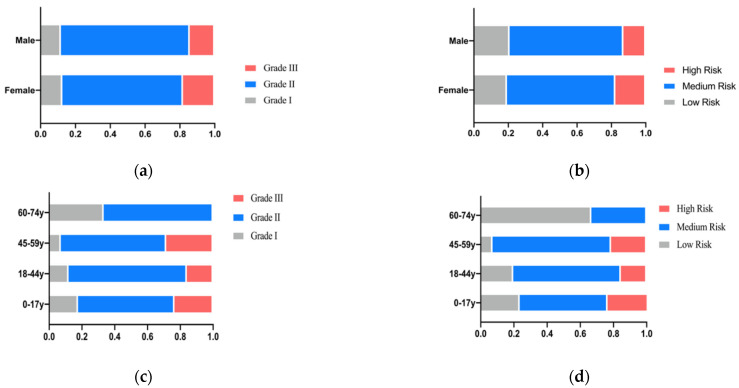
(**a**) Treatment difficulty grade proportion in gender; (**b**) Risk level proportion in gender; (**c**) Proportion of treatment difficulty grades in each age group; (**d**) Proportion of risk levels in each age group; (**e**) Proportion by occupation category; (**f**) Proportion of treatment difficulty grades in each occupational category; (**g**) Proportion of risk levels in each occupational category.

**Figure 11 ijerph-19-14069-f011:**
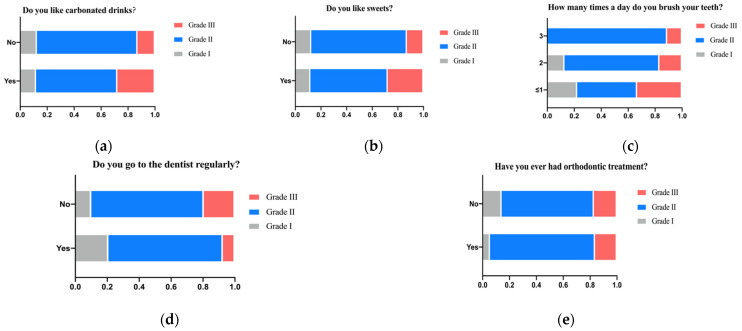
(**a**) Patients who like to drink carbonated drinks have a higher proportion of Grade III difficulty. (**b**) Patients who like sweets have a higher proportion of Grade III difficulty. (**c**) Patients who brush their teeth less than once a day have a higher proportion of Grade III difficulty. (**d**) Patients who go to the dentist regularly have a lower proportion of Grade III difficulty. (**e**) Patients who ever had orthodontic treatment have a lower proportion of Grade III difficulty.

**Table 1 ijerph-19-14069-t001:** The diagnosis and treatment of the patient.

Diagnosis	Treatment
Gingivitis	Periodontal basic treatment and dental health education
Dental caries (teeth #17–27, #37–47)	Permanent resin filling
Chronic pulpitis (tooth #46)	Root canal therapy

## Data Availability

Data are contained within the article.

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
