# Peer review of "Application of a Caries Treatment Difficulty Assessment System in Dental Caries Management"

_ijerph, 2022, doi:10.3390/ijerph192114069_

Round 1

Reviewer 1 Report (New Reviewer)

The paper entitled “Application of Caries Treatment Difficulty Assessment system in Dental Caries Management” is an interesting contribute that propose an original software useful for the assessment of applied caries risk and caries difficulty assessment. The authors showed a case followed up for five years, and some data collected using the software. The paper is original and of interest for the readers but requires some corrections before it can be considered valid for publication.

1. Although the authors cited their previous paper on "Caries difficulty assessment", specific assessment indicators can still be added to the Supplementary materials for the convenience of readers.

2. Names of bacteria mentioned on page 4 should be in italics.

3. Figure 7 on page 7 showed the secondary caries two years after treatment. The location of the secondary caries can be marked with a red arrow to facilitate reading.

4. The purpose of this case report is to demonstrate the application of the software, so the clinical relevance of the software should be explained more in the conclusion.

Author Response

Reviewer 2 Report (New Reviewer)

This is an interesting and informative article on the application of a PC software with the function of caries risk assessment and caries management planning. A clinical case with multiple caries lesions was managed with the assistance of the software. An analysis of the clinical records in the software system was performed. The followings are some suggestions to improve the quality of the article.

1.     Figure 1 showed how to use the software of the caries management system. Please give more information about what clinical information should be collected to use this system. The interface of the software should be clearly presented.

2.     A more detailed description of the clinical status of the patient should be given (2.2). Caries involved teeth, the severity of the caries lesion, and the pulpal status should be elaborated. Perforation was not accurate to describe Tooth 46.

3.     Please provide clinical photos showing the complete lower arch (Figure 2c)

4.     Line 107-111 present the diagnosis of the patient. Diagnosis 2 and 3 were repetitive

5.     Line 117-118 How do you determine the mutans streptococci and Lactobacillus level of the patient?

6.     Figure 3b 4, chlorhexidine was recommended in the management plan of the high caries risk patient. But current evidence did not support chlorhexidine for the management of caries. Please justify

7.     How do you determine the follow-up period of the patient?

8.     How do you determine the difficulty level of the treatment? What did GRADE I, II, and III represent, respectively? What was a GRADE A, B or C dentist?

9.     Section 2.3, Were there any criteria for the group of patients you include in the analysis?

Author Response

Reviewer 3 Report (New Reviewer)

The case chosen for the present report was poorly selected since it has multiple pathologies added.

In the document that I am sending you will find the pertinent observations.

Author Response

Reviewer 4 Report (New Reviewer)

This case report provides an interesting proprosal to patient records for caries risk factors. 

Suggest improvement in current dental terms for diagnosis and capturing more types of investigations which are fundamental for caries diagnosis including habits and diet.

Cognitive behaviour therapy should be part of patient treatment and not just restorations required otherwise relapse and return of dental disease is imminent.

Prevention and remineralisation is the key.

Reference and comparison with ICCMS and ICDAS is essential

Suggest improvements in capturing all of patient risk factors within the software program.

Round 2

Reviewer 2 Report (New Reviewer)

Thank you for addressing my questions!

Please revise Diagnosis 2 from 'soft drink dental caries' to 'dental caries' with the involved tooth number (Line 127 & Table 2). I have no other questions.

Congratulations on the successful development of the caries management software.

Author Response

We are very grateful to you for reviewing the paper so carefully.

We have revised Diagnosis 2 from 'soft drink dental caries' to 'dental caries' with the involved tooth number according to your suggestion.

Thanks for your kind reminders.

Reviewer 3 Report (New Reviewer)

In the 270 and 271 lines the  authors point out :

Our software is the first caries 270 management software that applied caries treatment difficulty assessment system.

 The authors insist that there are no programs that provide alternative care, as an example, we have the Cariogram that I have broken down and the Caries Risk Semaphore, which I did not present.

In this sense the CARIOGRAM programme specify in the:

Functions

By clicking on the icon button in the upper left corner of the screen you get information about the following functions:

Close, if you want to close the program

New, if you want to get a new empty screen (for a new patient)

About, to get information about the intellectual property of the program and about its use as an educational program.

Help, to get more information on how to run the program.

Notes, to record and write down your patient's comments.

 Preliminary interpretation, proposed preventive measures and clinical actions you could take, based on the data loaded into the program.

Print, to print the Cariogram and recommendations (Cariogram and recommendations).

 Clinical actions to be taken after obtaining the results.

Some suggestions for preventive actions can be found if you click on the '"Preliminary interpretation and proposed measures" icon in the upper left corner (see graphic below).

The main problem I detect is that by not identifying the risk and adjusting the pertinent preventive measures, as in the case presented, any treatment ultimately fails. 

Author Response

We are very grateful to you for reviewing the paper so carefully.

We agree with the opinion that there are many programs that applied the caries risk assessment, such as the CRA, CAMBRA and CARIOGRAM. The sentence in line 270 to 271 of our manuscript means that our software applied the Caries Treatment Difficulty Assessment System which had published in a previous study (Cheng, L., Zhang, L., Yue, L., Ling, J., Fan, M., Yang, D., Huang, Z., Niu, Y., Liu, J., Zhao, J., Li, Y., Guo, B., Chen, Z., & Zhou, X. (2022). Expert consensus on dental caries management. International journal of oral science, 14(1), 17. https://doi.org/10.1038/s41368-022-00167-3. ),

To avoid misunderstanding, we have revised this sentence as:

Our software applied not only the caries risk assessment, but also the Caries Treatment Difficulty Assessment. (Ln 306-307)

We agree with the opinion that “by not identifying the risk and adjusting the pertinent preventive measures, as in the case presented, any treatment ultimately fails.”

Our software also included the caries risk assessment of CAMBRA (Figure 3a, Figure6a) and gave personalized management plan (Figure 3b, Figure 6b).

Thanks for your kind reminders.

Reviewer 4 Report (New Reviewer)

Please review language and terminology as highlighted. Thank you for taking previous suggestions on board.

Author Response

We are very grateful to you for reviewing the paper so carefully.

We have carefully revised the highlights. Please see the attachment for details.

Thanks for your kind reminders.

This manuscript is a resubmission of an earlier submission. The following is a list of the peer review reports and author responses from that submission.

Round 1

Reviewer 1 Report

Manuscript title

Application of Caries Treatment Difficulty Assessment system 2 in Dental Caries Management

The authors present results from a study conducted in China to combine the well-applied caries risk assessment with caries difficulty assessment, to program and apply the Dental Caries Management Software, and analyze the 253 cases collected, to provide evidence for the improvement of caries management program, aiming to achieve better caries management effect. Finally, a case is presented to demonstrate the practical application of this software. This study could be of interest, but requires changes prior to publication.

Abstract

In general, the abstract is not well balanced.

- Too much information presented from the background before saying the objective.

- The objective of the study is not adequately presented; it is very broad and confusing. The verbs used in the objectives must be written in the infinitive. Consider making changes/rewriting it.

- The authors did not state the study design.

- Mention the setting, city and country where the study was conducted.

- Consider mentioning the age range of the participants.

- Did the authors perform any type of data analysis? (basic analysis?)

- Present the results for (average) age in the results and sex (percentage).

- Reconsider rewriting the conclusion based on the current study and research results

Introduction

- Consider the introduction classic 3-paragraph: What we know; what we no-know; and why this study was done. Please consider splitting the single paragraph of the introduction to make it more understandable to the reader.

- What is the research question and what is the hypothesis?

- The objective of the study is not adequately presented; it is very broad and confusing. The verbs used in the objectives must be written in the infinitive. Consider making changes/rewriting it.

- Explain the reasons and the scientific basis of the investigation. What is the rationale for the study?

Material and Methods

In general, the methodology needs to be described in greater detail to be accepted. What do the authors intend to present? What study design did you do? I believe that the authors should consult what should be reported according to the study carried out and resubmit their study, or series of cases.

To my knowledge, the authors made a scale and tied it to software, but it doesn't appear to be a scientific study.

- The authors did not state the study design. Please define.

- Mention the country where the study was conducted.

- Please provide additional information on inclusion and exclusion criteria.

- How did the authors arrive at the sample? Was there any sample size calculation?

- Also mention in this section the ethical aspects.

Results and Discussion

- Do the authors believe that it is necessary to mention the journal where what they want to comment on is published?

- This section is very difficult to follow. It does not appear to be a scientific article or the authors fail to convey what they want to say.

- what is the usefulness of the manuscript?

Reconsider rewriting

Conclusion

- The authors say nothing about their results. Conclusions shown are not conclusions derived from their results. It is necessary to add conclusions based on the results of your study. Reconsider rewriting the conclusion based on the current study and research results. Please elaborate. 

Reviewer 2 Report

The work should be improved with a better description of the development of the new method and protocol, because all that is stated are known things from clinical practice. It is necessary to test the effectiveness and purpose on a larger number of respondents. It is not clear what the application would contribute at a wider level in the general population.

Reviewer 3 Report

The paper entitled “Application of Caries Treatment Difficulty Assessment system  in Dental Caries Management” is an interesting contribute that propose an original software usefull for the assessment of applied caries risk and caries difficulty assessment. The paper is original and of  interest for the readers but requires some corrections before it can be considered valid for publication.

INTRODUCTION

Overall well structured, even if too synthetic; authors should provide all the information necessary to understand the scientific background, the knowledge gap and the objectives of the study.

MATERIAL AND METHODS

The development process and how the software works are described in too general a manner. There is a lack of adequate iconography illustrating the different working panels of the software and its characteristics, as the image in Fig. 1 is only explanatory of the information flow mode. More clear and explanatory figures should be added in this section in order to enable the reader to understand the real functioning of the original software developed in this study. The figure on page 10 incorrectly numbered as Fig. 1, as in sequence it should be numbered as Fig. 8, shows a patient card; however, it would be more correct to include these  informations in the materials and methods section, explaining more fully the criteria used for assessing the risk of caries and caries treatment difficulty

RESULTS AND DISCUSSION

The results section and the discussion of the results should be reported separately so that readers can better understand the results found by the authors in the study and compare them with the relevant scientific evidence in the international literature.

Clinical relevance of the use of softwares such the original one proposed in the study should be emphasized more, in order to provide suggestions to the clinician usefull in daily practice.

CONCLUSION

Conclusions are limited to a summary of the results obtained; considering the clinical relevance of the topic, it is necessary to insert a clear take home message that can be useful to modify or to adapt the reader's practice.

As structured, the conclusion section appears to be a concluding abstract of the work; this section needs to be completely revised and report, preferably with a bulleted list, only the key results of the study.